# KNOW2BIO: A COMPREHENSIVE DUAL-VIEW BENCHMARK FOR EVOLVING BIOMEDICAL KNOWLEDGE GRAPHS

## ABSTRACT

Knowledge graphs (KGs) have emerged as a powerful framework for representing and integrating complex biomedical information. However, assembling KGs from diverse sources remains a significant challenge in several aspects, including entity alignment, scalability, and the need for continuous updates to keep pace with scientific advancements. Moreover, the representative power of KGs is often limited by the scarcity of multi-modal data integration. To overcome these challenges, we propose Know2BIO, a general-purpose heterogeneous KG benchmark for the biomedical domain. Know2BIO integrates data from 30 diverse sources, capturing intricate relationships across 11 biomedical categories. It currently consists of ~219,000 nodes and ~6,200,000 edges. Know2BIO is capable of user-directed automated updating to reflect the latest knowledge in biomedical science. Furthermore, Know2BIO is accompanied by multi-modal data: node features including text descriptions, protein and compound sequences and structures, enabling the utilization of emerging natural language processing methods and multi-modal data integration strategies. We evaluate KG representation models on Know2BIO, demonstrating its effectiveness as a benchmark for KG representation learning in the biomedical field. Data and source code of Know2BIO are available at https://anonymous.4open.science/r/Know2BIO/.

## 1 INTRODUCTION

A knowledge graph (KG), represents entities as nodes and their relations as edges, commonly referred to as "triples" (h, r, t), where a head entity (h) is connected to a tail entity (t) by a relation (r). There is an increasing presence of using KGs to represent the data in knowledge bases (KBs). In biomedical science, KBs capture knowledge in domains such as omics (e.g., genomics (Cunningham et al. (2021); Seal et al. (2022); O'Leary et al. (2015); Sayers et al. (2021)), proteomics (Bateman et al. (2022); Fabregat et al. (2013); Szklarczyk et al. (2018)), metabolomics (Powell & Moseley (2022); Jewison et al. (2013); Wishart et al. (2021))), pharmacology (e.g., drug designs (Wishart et al. (2017); Kanehisa et al. (2016); Davis et al. (2022), drug targets Wishart et al. (2017); Zhou et al. (2021)), adverse effects (Giangreco & Tatonetti (2021); Kuhn et al. (2015))), physiology (e.g., biological processes (Carbon et al. (2020); Kanehisa et al. (2016); Fabregat et al. (2013)), and anatomical components (Haendel et al. (2014); Mungall et al. (2012); Lipscomb (2000); Bastian et al. (2020)), playing a vital role in advancing biomedical research and data science .

This knowledge has been employed by predictive algorithms to discover new biomedical knowledge (e.g., protein interactions, pathogenic genetic variants). To enable this knowledge discovery, task-relevant data must be integrated from multiple sources such as drug-relevant data in KGs for predicting drug targets (Ioannidis et al. (2020); Yan et al. (2021); Mayers et al. (2022); Himmelstein et al. (2017); Zong et al. (2022); Su et al. (2023)) and clinically-relevant data in KGs to predict clinical characteristics indicative of pathogenesis (Santos et al. (2022); Gao et al. (2022); Chandak et al. (2023); Liang et al. (2022)). However, this data integration has posed challenges, resulting in KGs which insufficiently represent biomedicine, are unsuited for new tasks, and do not keep pace with biomedical advancements.

Predictive algorithms for KGs include KG representation learning models. These models learn low-dimensional embeddings to capture the contextual information of entities and their relationships. Existing models can be categorized into five main types: 1) Translation-based models represent relations between entities as translations in the embedding space (Bordes et al. (2013); Wang et al. (2014); Lin et al. (2015); Ji et al. (2015)). 2) Bilinear models utilize bilinear forms to capture the interactions between entities and relations in the embedding space (Yang et al. (2014); Kazemi & Poole (2018)). 3) Neural network-based models utilize deep neural networks to learn representations of entities and relations (Socher et al. (2013); Dong et al. (2014); Dettmers et al. (2017); Nguyen et al. (2017)). 4) Complex vector-based models utilize complex vector spaces (Trouillon et al. (2016); Sun et al. (2018); Chami et al. (2020)). Lastly, 5) Hyperbolic space embedding models utilize hyperbolic space which represents hierarchical structures with minimal distortion (Balazevic et al. (2019); Chami et al. (2020)). Each of these model categories are detailed in the Appendix B.

Biomedical KG construction demands several technical considerations: (1) *Entity representation*: Different knowledge sources may represent the same entities differently necessitating accurate alignment to avoid redundancy and false information (Zong et al. (2022)). (2) *Continuous knowledge updates:* Biomedical science evolves rapidly. As a result, the one-time efforts to assemble a KG can quickly fall behind the latest biomedical knowledge, hindering biomedical discovery and real-world benchmarking. Thus, it is essential to establish mechanisms to keep the KG up-to-date. (3) *Representative power:* Although biomedical KGs are inherently incomplete due to gaps in biomedical knowledge, existing KGs fail to capture known biomedical knowledge. Furthermore, these KGs are scarcely supplemented with other data modalities such as molecular sequences, molecular structures, or natural language descriptors which can be combined with other representation learning methods such as language models.

Therefore, we propose a comprehensive and evolving general-purpose KG: Knowledge Graph Benchmark of Biomedical Instances and Ontologies (Know2BIO). Know2BIO represents the biomedical domain more comprehensively than popular biomedical knowledge graph benchmarks; it is larger (219,000 nodes, 6,180,000 edges), integrates data from more sources (30 sources), represents 11 biomedical categories, and includes biomedically-relevant edge types not present in other KGs (e.g., anatomy-specific gene expression, transcription factor regulation of genes). Not only is its data more up-to-date, but unlike others, it can be automatically updated to reflect the most recent biomedical knowledge obtained from its data sources. By representing the latest scientific knowledge, Know2BIO defines a better real world learning task for graph learning methods and provides a greater opportunity for biomedical knowledge discovery. Additionally, Know2BIO enables methods development at the forefront of graph learning: its instance and ontology views enable multi-view learning tasks; its multi-modal node features (e.g., natural language descriptors, chemical sequences, protein structures) enable multi-modal learning and data integration strategies (Wan et al. (2018); Zong et al. (2019); Luo et al. (2017); Huang et al. (2020)), as well as advanced NLP techniques such as language models (Huang et al. (2019); Lee et al. (2019); Rives et al. (2019); Heinzinger et al. (2019)). By providing a comprehensive KG that can reflect—in perpetuity—the latest biomedical knowledge, Know2BIO serves as an excellent benchmark to evaluate a variety of KG representation learning models under various scenarios (e.g., biomedical use cases, ablation studies, multi-modal data).

We extensively evaluate 13 KG representation models from the 5 aforementioned categories on a KG-wide link prediction task (predicting missing nodes in triples). We find that the complex and hyperbolic models perform better than translation and bilinear models in the ontology view and to a lesser extend in the instance view due to its greater denser and diversity. Our contributions are as follows:

- Know2BIO is a general purpose heterogeneous KG representing a diverse array of informative biomedical categories covering real-world data

- Know2BIO can be automatically updated, reflecting the latest biomedical knowledge

- Know2BIO enables multi-modal learning strategies by including node features such as natural language text descriptors; sequences for proteins, compounds, and genes; structures for proteins and compounds.

- Know2BIO enables multi-view learning by including and specifying two views of the KG

- Benchmarking of KG representation learning methods is performed on our KG across 13 different models for 3 spaces: Euclidean, complex, and hyperbolic.

## 2 RELATED WORKS

### 2.1 BIOMEDICAL KNOWLEDGE GRAPHS

Several biomedical KGs have been released in recent years. **Hetionet** ((Himmelstein et al. (2017)) has been applied to predict disease-associated genes and for drug repurposing but is now relatively small and less up-to-date. Amazon's **DRKG** (Ioannidis et al. (2020)), has twice as many nodes, though it has has a narrow focus on COVID-19 drug repurposing. The Mayo Clinic's **BETA** (Zong et al. (2022) )is a benchmark for predicting drug targets, but it is largely composed of older data from Bio2RDF (Belleau et al. (2008)) and its size is quite inflated due to unaligned nodes. **PharmKG** (Zheng et al. (2020)) includes non-graph data modalities for node features (e.g., gene expression, disease word embeddings), but it is relatively small and only has 3 node types (Zheng et al. (2020)). The **iBKH** KG (Su et al. (2023)) represents the general biomedical domain, and although it is larger, over 90% of its nodes are molecule nodes linked to drug compounds. **CKG** (Santos et al. (2020)) is a massive KG for clinical decision support, integrating experimental data, publications, and biomedical KBs; however, the text mined data potentially introduce additional uncertainty, compared to carefully curated findings from biomedical KBs and its size may be intractable. **Open Graph Benchmark (OGB)** (Hu et al. (2020)), a collection of KG benchmarks has a biomedical KG, ogbl-biokg, but it only includes 5 biomedical categories and is limited in size. Although **OpenBioLink** (Breit et al. (2019)), is large and high-quality and was intended to be updated, but like all other such KGs benchmarks, it has not been continually updated. The **COVID-19 KG** (Wise et al. (2020)) introduces a heterogeneous graph for visualizing complex relations in COVID-19 literature. **HKG-DDIE** (Asada et al. (2022)) proposes a method to extract drug-drug interactions by integrating diverse pharmaceutical KG data with corpus text and drug information.

In sum, incomplete entity alignment, restricted focuses, and data that is uni-modal and single-view hamper existing biomedical KG utility for real-world benchmarking and biomedical discovery. Table 1 summarizes statistics of these KGs together with the Know2BIO proposed in this paper.

Table 1: An overview of heterogeneous biomedical KG

| Dataset | #Entities (millions) | #Relations (millions) | #Node types | #Edge types | #Source databases |
|---------|---------------------|----------------------|-------------|-------------|-------------------|
| BETA | 0.95 | 2.56 | 3 | 9 | 9 |
| CKG | 16.0 | 220.0 | 36 | 47 | 15 |
| DRKG | 0.097 | 5.87 | 13 | 17 | 6 |
| Hetionet | 0.047 | 2.25 | 11 | 24 | 29 |
| iBKH | 2.38 | 48.19 | 11 | 18 | 17 |
| OGB:biokg | 0.093 | 5.09 | 5 | 6 | / |
| OpenBioLink | 0.184 | 9.30 | 7 | 30 | 16 |
| PharmKG | 0.188 | 1.09 | 3 | 29 | 6 |
| COVID-19 KG | 0.336 | 3.33 | 5 | 5 | 1 |
| HKG-DDIE | 0.021 | 2.75 | 5 | 8 | 5 |
| **Know2BIO** | 0.219 | 6.18 | 16 | 108 | 30 |

### 2.2 KNOWLEDGE GRAPH BENCHMARKING

There have been several widely used general domain KG benchmarks that propelled the development of many KG representation learning models. One of the most widely-used KG benchmarks is **FB15K**(Bordes et al. (2013), a dense general purpose KG derived from Freebase. **YAGO** (Tanon et al. (2020) is another widely used high-quality KB covering general Wikipedia-derived ontological and instance knowledge about people, places, movies, and organizations. **DBpedia** (Lehmann et al. (2015)) is a similar popular KG of Wikipedia data. **CoDEx** (Safavi & Koutra (2020)) also uses Wikidata and Wikipedia data for link prediction. Other benchmark initiatives such as **OGB** and **TUDataset** host multiple benchmark datasets from various domains and of various scales (Hu et al. (2020; 2021); Morris et al. (2020)). They cover citation networks, commercial products, small molecules, bioinformatics, social networks, and computer vision. Although these KGs can work well

in their own domain, performance on non-biomedical data often does not generalize to the biomedical domain. To address challenges from biomedical science, rigorous general purpose biomedical KG benchmarks must be employed, motivating Know2BIO.

## 3 KNOW2BIO KNOWLEDGE GRAPH

We propose a general-purpose biomedical KG, Know2BIO which represents 11 biomedical categories across 16 node types, totaling 219,169 nodes, 6,181,160 edges, and 30 unique pairings of node types across 108 unique edge types. Most node pairs have 1-2 edge types, while compound-to-protein edges have 51 unique edge types. Compound-to-compound edges are the most numerous, at 2,902,659 edges.

Table 2: Scale and average degree of each biomedical category.

| Biomedical Category | Total nodes | Total edges | Average node degree |
|---|---|---|---|
| Anatomy | 4,960 | 226,630 | 45.7 |
| Biological Process | 27,991 | 209,959 | 7.5 |
| Cellular Component | 4,096 | 96,239 | 23.5 |
| Disease | 21,842 | 419,338 | 19.2 |
| Compound | 26,549 | 3,561,235 | 134.1 |
| Drug Class | 5,721 | 10,859 | 1.9 |
| Gene | 28,476 | 1,757,428 | 61.7 |
| Molecular Function | 11,272 | 85,779 | 7.6 |
| Pathway | 52,215 | 467,420 | 9.0 |
| Protein | 21,879 | 1,937,114 | 88.5 |
| Reaction | 14,168 | 236,113 | 16.7 |
| Total | 219,169 | 6,181,160 | - |

Node features, i.e., data from additional modalities, are provided separate from the KG, enabling users to integrate and embed such data with different models and feature fusion strategies of their choosing. These node features include DNA sequences for ~22,000 gene nodes, amino acid sequences for ~21,000 protein nodes, the SMILES sequence of ~7,200 compound nodes (sequences which can be turned into graphs/structures), structures for ~21,000 protein nodes, and text descriptors for ~208,500 nodes.

### 3.1 KNOWLEDGE GRAPH CONSTRUCTION

To construct our KG, we integrate data from 30 data sources spanning several biomedical disciplines (Table 3, Appendix A). We carefully selected data sources and aligned the provided data. Alignment entailed mapping data identifiers (IDs) to common IDs through various intermediary resources. This is critical because data sources frequently use different IDs to represent the same entity (e.g., gene IDs from NCBI/Entrez, Ensembl, or HGNC). However, this process can be circuitous. For example, to unify knowledge on compounds and the proteins they target (i.e., *Compound (DrugBank ID) -targets- Protein (UniProt ID)*) taken from the Therapeutic Target Database (TTD), the following relationships are aligned: *Compound (TTD ID) -targets- Protein (TTD ID)* from TTD, *Protein (TTD ID) -is- Protein (UniProt name)* from UniProt, and *Protein (UniProt name) -is- Protein (UniProt ID)* from UniProt. This creates *Compound (TTD ID) -targets- Protein (UniProt ID)* edges. But to unify this with the same compounds represented by DrugBank IDs elsewhere in the KG, the following relationships are aligned: *Compound (DrugBank ID) -is- Compounds (old TTD, CAS, PubChem, and ChEBI IDs)* from DrugBank (4 relationships), and *Compounds (CAS, PubChem, and ChEBI) -is- Compound (new TTD)* from TTD (3 relationships). Appendix C and out GitHub[1] provide details on Know2BIO's unique relations between entity types.

Relationships are also backed by varying levels of evidence (e.g., for STRING's protein-protein associations and DisGeNET's gene-disease associations). To select appropriate evidence requirements for inclusion in our KG, we investigated how confidence scores are calculated, what past researchers have selected, KB author recommendations, and resulting data availability[2]. Many manually-curated

---

[1] `https://anonymous.4open.science/r/Know2BIO/dataset/create_edge_files_utils`

[2] `https://anonymous.4open.science/r/Know2BIO/dataset/create_edge_files_utils/README.md`

sources did not provide confidence scores (e.g., GO, DrugBank, Reactome) and are ostensibly high-confidence sources which were not filtered by confidence.

Table 3: Data Sources for Know2BIO's Biomedical Categories

| Biomedical Category | # Data Sources | Data Sources | Original Identifiers | Identifier(s) Aligned To |
|---|---|---|---|---|
| Anatomy | 4 | Bgee (Bastian et al. (2020), PubMed, MeSH(Lipscomb (2000)), Uberon (Haendel et al. (2014); Mungall et al. (2012)) | MeSH ID, MeSH tree number | MeSH ID, MeSH tree number |
| Biological process | 1 | GO (Carbon et al. (2020); Ashburner et al. (2000)) | GO | GO |
| Cellular component | 1 | GO | GO | GO |
| Compounds/Drugs | 11 | DrugBank (Wishart et al. (2017)), MeSH, CTD (Davis et al. (2022)), UMLS (Bodenreider (2004)), KEGG (Kanehisa et al. (2016)), TTD (Zhou et al. (2021)), Inxight Drugs (Siramshetty et al. (2021)), Hetionet (Zhu et al. (2019)), PathFX (Wilson et al. (2018)), SIDER (Kuhn et al. (2015)), MyChem.info (Lelong et al. (2021)) | DrugBank, MeSH ID, UMLS, UNII, ATC, KEGG Drug, KEGG Compound, PubChem Substance (Kim et al. (2022)), PubChem Compound (Kim et al. (2022)), CAS (Jacobs et al. (2022)), InChI (Heller et al. (2015)), SMILES (Weininger (1988)), ChEBI (Hastings et al. (2015)), TTD (two versions) | DrugBank, MeSH ID |
| Disease | 14 | PubMed, MeSH, DisGeNET(Piñero et al. (2021)), SIDER, ClinVar(Landrum et al. (2019)), ClinGen (Rehm et al. (2015)), PharmGKB(Gong et al. (2021)), MyDisease.info(Lelong et al. (2021)) PathFX, UMLS, OMIM, Mondo, DOID(Schriml et al. (2021)), KEGG | MeSH ID, MeSH tree number, UMLS, DOID, KEGG, OMIM(Amberger et al. (2018)), Mondo(Vasilevsky et al. (2022)) | MeSH ID, MeSH tree number |
| Drug Class | 1 | ATC | ATC | ATC |
| Genes | 9 | HGNC, GRNdb (Fang et al. (2020)), KEGG, ClinVar, ClinGen, SMPDB (Jewison et al. (2013)) DisGeNET (Piñero et al. (2021)), PharmGKB (Gong et al. (2021)), MyGene.info (Lelong et al. (2021)) | Entrez, Ensembl (Cunningham et al. (2021)), HGNC (Seal et al. (2022)),Gene name | Entrez |
| Molecular function | 1 | GO | GO | GO |
| Pathways | 3 | Reactome(Fabregat et al. (2013)), KEGG, SMPDB | Reactome, KEGG, SMPDB | Reactome, KEGG, SMPDB |
| Proteins | 6 | UniProt (Dogan (2018); Bateman et al. (2022)), Reactome, TTD SMPDB, STRING, HGNC | UniProt, STRING (Szklarczyk et al. (2018)), TTD | UniProt |
| Reactions | 1 | Reactome | Reactome | Reactome |

The discrepancy between the number of biomedical categories (11) and node types (16) was due to complexities in the data identifiers. There are two node types for compounds, DrugBank IDs and MeSH IDs, because an incomplete amount of such identifiers could be aligned. (Overall, the compound identifier alignment process was the most arduous mapping.) Instead of merging the aligned nodes and discarding the significant number of unaligned nodes, we retained the two node types, mapping ~nine other compound identifiers to those two. There are three node types for biological pathways because, after attempting to align pathways (e.g., via comparing pathways' genes, proteins, and names), pathways from the three pathway ontologies could not be aligned—even by loose definitions. This is understandable because pathway definitions are partially subjective based on the human biocurators' focuses (e.g., SMPDB focuses on small molecule drug pathways). There are two node types for anatomy. One node type, MeSH ID, is an instance of an anatomy which could be categorized under multiple branches in an ontology, while the other, MeSH tree number, is an anatomical category unique to one point in an ontology. There are two such node types for disease as well, for the same reason and of the same identifiers. They were employed to take advantage of the instance and ontology view.

Despite the arduous nature of integrating the data, users can easily run the scripts we provide on our GitHub[3] to automatically obtain and integrate the data from the latest versions of the data sources. (Note that due to access requirements, users must create free accounts for DrugBank, UMLS, and DisGeNET, and then manually download two files into the input folder. After that, all scripts can be run to obtain and integrate data from these and the ~27 other sources.)

## 3.2 DUAL VIEW KNOWLEDGE GRAPH

Often, a node in a KG may represent an entity in the *instance view* (e.g., a specific compound such as ibuprofen) or a concept in the *ontology view* (e.g., a compound category such as cardiovascular system drugs). The relations in the instance view can be interactions, associations, and other edges that relate objects to one another. Relations in the ontology view are typically hierarchical and relate how one concept is a sub-concept of another. Although jointly learning embeddings of these separate views can inform and improve performance on downstream tasks such as link prediction (Hao et al. (2019); Iyer et al. (2022); Hao et al. (2020)), most KG representation learning methods fail to take advantage of this potential.

To enable multi-view learning, Know2BIO includes both views. For example, the instance view includes edges describing protein-drug interactions, while the ontology view includes functional information for proteins (e.g., pathway ontologies). The two views are connected by bridge nodes. Together, the two views and bridge nodes form the whole view of the KG (Figure 1).

---

[3]https://anonymous.4open.science/r/Know2BIO/dataset

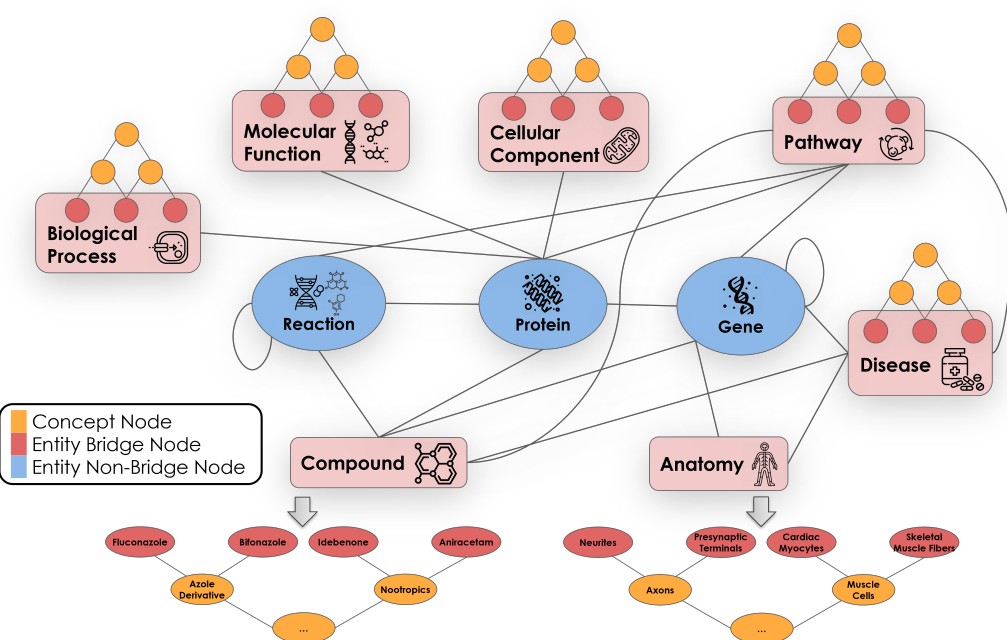

Figure 1: Schema of Know2BIO.

Blue nodes represent instance entities, red nodes represent bridge entities, and orange nodes represent concepts. The orange nodes are structured in a hierarchy to represent the ontology (i.e., concept hierarchies).

## 4 BENCHMARKING KNOW2BIO

### 4.1 DATASETS

We comprehensively evaluate Know2BIO from thtee views: ontology, instance, and whole views. Bridge nodes connect the ontology and instance views and are only evaluated as part of the whole view. The resulting KG from Section 3 was split into a train and test KG using the GraPE package (Cappelletti et al. (2021)). All connected components with greater than 10 nodes were included in the train/test/validation split. To ensure connectivity of the train KG, the training set included the minimum spanning tree of each component, with up to 20% of remaining edges split evenly between test and validation. Table 4 provides the exact data splits.

Table 4: Summary statistics of Know2BIO 's different views: number of nodes, relation types, training set triples, validation set triples, test set triples, and total triples

|  | Number of entities | Type of relations | Train | Valid | Test | Total |
|---|---|---|---|---|---|---|
| **Ontology** | 68,314 | 5 | 93,056 | 8,368 | 8,367 | 109,827 |
| **Bridge** | 102,111 | 29 | 366,780 | 45,748 | 45,748 | 475,779 |
| **Instance** | 145,445 | 76 | 3,320,385 | 415,050 | 415,050 | 5,595,554 |
| **Whole** | 219,169 | 108 | 3,780,221 | 469,166 | 469,165 | 6,181,160 |

### 4.2 EXPERIMENTS

**Evaluation Tasks**   To failry compare and benchmark different types of models on Know2BIO, we adopt the commonly used link prediction task. This is the task of predicting a missing node (h/t) in a triple (h, r, t). Given the potentially vast number of entities in a KG, simply predicting a single most likely candidate does not provide a comprehensive evaluation metric. Consequently, models typically rank a set of candidate nodes in the KG. For each test triple (h, r, t), h/t is substituted with candidate entities in the KG. The model computes the scores of candidate entities and ranks them in descending order. We employ the Hits@k and MRR (mean reciprocal rank) evaluation metrics.

Hits@k quantifies the proportion of correct entities that are present within the initial $k$ entities of the sorted rank list. MRR computes the arithmetic mean of the reciprocal ranks.

**Experiment Setup**   As hyperparameter tuning has been demonstrated to strongly impact model performance and enable fair comparisons between models, (Bonner et al. (2021)). we performed hyperparameter tuning with beam search on the batch size (512, 1024, 2048), learning rate ($1e^{-4}$,$5e^{-4}$,$1e^{-3}$,$1e^{-2}$,$1e^{-1}$), and negative sampling ratio (None, 5, 25, 50, 100, 125, 150, 250). We fix the maximum training epoch to be 1000 and early stopping patients to 5 epochs. Early stopping ensures that the models are sufficiently trained and avoids overfitting. Negative samples are constructed by replacing a positive triple's tail with a random node from the entire knowledge graph. We utilize the Adam optimizer (Kingma & Ba (2014)) for Euclidean and hyperbolic models and SparseAdam for complex space models. SparseAdam is an Adam variant designed to handle sparse gradients and work efficiently with dense parameters. For testing, metrics are calculated by averaging the prediction performances on both heads and tails. Predictions are filtered by edge types' respective node types. All the models' hidden sizes are set to 512 to ensure a fair comparison. All experiments are performed on 2 servers with AMD EPYC 7543 Processor (128 cores), 503 GB RAM, and 4 NVIDIA A100-SXM4-80GB GPUs. To ensure reproducibility and make Know2BIO accessible to the computer science and biomedical community, complete training, validation, and testing configuration files are available in Know2BIO's repository[4].

## 4.3   RESULTS

We benchmark Know2BIO's ontology, instance, and whole views—not just a single view (Chang et al. (2020))—with models from Euclidean, complex, and hyperbolic spaces. Models are categorized into complex space, hyperbolic space, and Euclidean space models. Euclidean space models are further categorized into distance-based (Euclidean distance similarity) and semantic-based (dot similarity) models. For researchers unfamiliar with models' mechanisms, in Table 12 we detail their scoring functions (Nguyen (2020); Ji et al. (2020)).

**Ontology View**   The ontology view of Know2BIO is characterized by a tree-like structure with 5 relation types, much less than the 76 types in the more densely connected instance view (Table 4). This scarcity of neighboring information makes modeling Know2BIO's ontology view non-trivial. Such properties enable researchers to better evaluate their models' capacity to capture biomedical knowledge in a hierarchical manner. Such hierarchical relations are best modeled by hyperbolic space models(Chami et al. (2020)) which outperform Euclidean and complex space models on average (Table 5, 6).

Table 5: Ontology View: Euclidean Space

| Ontology View | | | | | | |
|---|---|---|---|---|---|---|
| Category | Model | Performance | | | | |
| | | MR | MRR | Hit@1 | Hit@3 | Hit@10 |
| **Distance** | TransE | 1323.58 | 0.0799 | 0.0186 | 0.0743 | 0.2103 |
| | TransR | 1804.35 | 0.0813 | 0.0208 | 0.0746 | 0.2086 |
| | AttE | 2038.19 | 0.2120 | 0.1302 | 0.2344 | 0.3799 |
| | RefE | 1417.40 | 0.1836 | 0.1020 | 0.2013 | 0.3517 |
| | RotE | 2174.07 | 0.2143 | 0.1343 | 0.2382 | 0.3755 |
| | MurE | 1684.75 | 0.2094 | 0.1279 | 0.2310 | 0.3765 |
| **Semantic** | CP | 6658.02 | 0.1499 | 0.0693 | 0.1692 | 0.3237 |
| | DistMult | 6706.65 | 0.1520 | 0.0690 | 0.1747 | 0.3334 |

**Instance View**   Know2BIO's instance view is more densely connected than the ontology view, providing more information for a node's embedding. However, enhanced context advantages also come at a price: the KG models need to represent more types of relations. Such properties enable researchers to better evaluate their models' capacity to capture the complex relations and structures in

---
[4]https://anonymous.4open.science/r/Know2BIO/benchmark/configs

Table 6: Ontology View: Complex and Hyperbolic Space

| Ontology View | | | | | | |
|---|---|---|---|---|---|---|
| **Category** | **Model** | **Performance** | | | | |
| | | MR | MRR | Hit@1 | Hit@3 | Hit@10 |
| **Complex** | RotatE | 8703.68 | 0.1061 | 0.0580 | 0.1202 | 0.2022 |
| | ComplEx | 9395.01 | 0.1342 | 0.0738 | 0.1504 | 0.2615 |
| **Hyperbolic** | AttH | 2151.64 | 0.2087 | 0.1253 | 0.2337 | 0.3788 |
| | RefH | 1372.03 | 0.1801 | 0.0962 | 0.1989 | 0.3522 |
| | RotH | 2272.63 | 0.2095 | 0.1287 | 0.2332 | 0.3722 |

biomedical knowledge graphs. Such relations are best modeled by the complex space models which outperform Euclidean and hyperbolic space models on average (Table 7, 8).

Table 7: Instance View: Euclidean Space

| Instance View | | | | | | |
|---|---|---|---|---|---|---|
| **Category** | **Model** | **Performance** | | | | |
| | | MR | MRR | Hit@1 | Hit@3 | Hit@10 |
| **Distance** | TransE | 1316.30 | 0.1171 | 0.0621 | 0.1259 | 0.2194 |
| | TransR | 1299.94 | 0.1233 | 0.0728 | 0.1275 | 0.2218 |
| | AttE | 725.61 | 0.1989 | 0.1400 | 0.2099 | 0.3116 |
| | RefE | 792.09 | 0.1792 | 0.1233 | 0.1881 | 0.2841 |
| | RotE | 794.64 | 0.1812 | 0.1250 | 0.1907 | 0.2871 |
| | MurE | 783.26 | 0.1946 | 0.1372 | 0.2050 | 0.3028 |
| **Semantic** | CP | 1427.38 | 0.0953 | 0.0481 | 0.0983 | 0.1827 |
| | DistMult | 1434.64 | 0.0968 | 0.0499 | 0.0995 | 0.1832 |

Table 8: Instance View: Complex and Hyperbolic Space

| Instance View | | | | | | |
|---|---|---|---|---|---|---|
| **Category** | **Model** | **Performance** | | | | |
| | | MR | MRR | Hit@1 | Hit@3 | Hit@10 |
| **Complex** | RotatE | 1178.16 | 0.2157 | 0.1410 | 0.2337 | 0.3662 |
| | ComplEx | 1601.89 | 0.1859 | 0.1131 | 0.2000 | 0.3335 |
| **Hyperbolic** | AttH | 841.83 | 0.1813 | 0.1250 | 0.1915 | 0.2872 |
| | RefH | 859.59 | 0.1712 | 0.1173 | 0.1787 | 0.2728 |
| | RotH | 874.41 | 0.1661 | 0.1112 | 0.1747 | 0.2687 |

**Whole View**  In the ontology view, hyperbolic models perform best. In the instance view, complex models perform best. Euclidean models lie in the middle on average, depending on the embedding transformation strategy. To provide a balanced benchmarking scheme, we have created the whole view by adding bridge nodes (Table 4), entities that connect the instance view to the ontology view nodes. Table 9 and Table 10 show the evaluation of the whole view. For researchers using Know2BIO, we recommend evaluation on at least the whole view, since it measures models' capacities to capture both conceptual knowledge (ontology view) and factual knowledge (instance view).

## 5    CONCLUSIONS

We have constructed and released a heterogeneous biomedical KG known as Know2BIO. This KG integrates information across 30 biomedical KBs, totaling over 219,000 nodes in 11 biomedical categories and 6,180,000 relationships. We evaluated representative KG models from Euclidean, complex, and hyperbolic spaces, providing a performance benchmark for future models on Know2BIO. Furthermore, we have developed an open-source framework for generating and updating a general

Table 9: Whole View: Euclidean Space

| Category | Model | Performance | | | | |
|---|---|---|---|---|---|---|
| | | MR | MRR | Hit@1 | Hit@3 | Hit@10 |
| Distance | TransE | 1508.11 | 0.1008 | 0.0545 | 0.1063 | 0.1839 |
| | TransR | 1542.63 | 0.1087 | 0.0657 | 0.1108 | 0.1907 |
| | AttE | 805.07 | 0.1677 | 0.1119 | 0.1766 | 0.2741 |
| | RefE | 854.55 | 0.1543 | 0.1027 | 0.1602 | 0.2513 |
| | RotE | 857.33 | 0.1568 | 0.1051 | 0.1629 | 0.2535 |
| | MurE | 846.02 | 0.1697 | 0.1154 | 0.1781 | 0.2717 |
| Semantic | CP | 1594.40 | 0.0952 | 0.0483 | 0.0983 | 0.1827 |
| | DistMult | 1584.33 | 0.0930 | 0.0451 | 0.0965 | 0.1823 |

Table 10: Whole View: Complex and Hyperbolic Space

| Category | Model | Performance | | | | |
|---|---|---|---|---|---|---|
| | | MR | MRR | Hit@1 | Hit@3 | Hit@10 |
| Complex | RotatE | 2639.09 | 0.1818 | 0.1166 | 0.1943 | 0.3128 |
| | ComplEx | 3419.68 | 0.1516 | 0.0857 | 0.1627 | 0.2832 |
| Hyperbolic | AttH | 973.13 | 0.1497 | 0.0969 | 0.1571 | 0.2498 |
| | RefH | 1012.54 | 0.1333 | 0.0855 | 0.1372 | 0.2223 |
| | RotH | 1004.64 | 0.1316 | 0.0830 | 0.1358 | 0.2221 |

biomedical KG which can be applied to answer biomedical research questions, such as drug development and therapeutics as well as disease biomarker discovery and prognosis. This framework is both scalable and extensible to allow for the integration of additional biomedical KBs. As the source databases update, researchers can use this framework to integrate the latest findings and create their own KGs. We will periodically update and release Know2BIO.

**Limitations:** Biomedical knowledge representation is inherently incomplete because biological systems are only partially understood. Incomplete knowledge of different biomedical data types can bias the data, resulting in different results over time as databases update, as has been shown with data from GO (Tomczak et al. (2018)). Although we sought evidence-backed reasons when choosing the confidence thresholds (Section 3.1), there is some arbitrariness. In this benchmark, the unweighted version of the graph was used, (e.g., equating all non-zero disease-disease similarity edges), which likely hindered the performance of some representation learning models.

**Future Work:** Future benchmarking work on Know2BIO can test various graph learning models (e.g., KG embeddings, GNNs), multi-modal models (e.g., text and graph embedding models), and training strategies (e.g., parameter initialization, curriculum learning). Multi-view graph learning approaches can use the instance and ontology views. Multi-modal models can use the non-KG data to further enrich the KG representation; non-KG data modalities can be extracted (e.g., by language models, graph learning models) and integrated as node information into KG learning models via different feature fusion approaches (e.g., early, mixed, late fusion). Some feature fusion approaches can also double as a pre-training parameter initialization strategy, one of many initialization strategies to test. The edge weights can be utilized by graph learning approaches that incorporate weights into the training objective. Edge weights can also be used for curriculum learning. This will enable researchers to access a more extensive and holistic view of biomedical knowledge, providing a wider range of benchmarking tasks and challenges.

Additionally, other emerging technologies can take advantage of Know2BIO. In particular, large language models (LLMs) can leverage the up-to-date information and structure of Know2BIO. This can facilitate reasoning and mitigate hallucination for the LLM, both for predictive analyses (e.g., in-context learning) and retrieval augmented generation (e.g., multi-hop question answering). We plan to explore each of these topics in future works.

ETHICS STATEMENT

Every author involved in this manuscript has reviewed and committed to adhering to the ICLR Code of Ethics.

REPRODUCIBILITY STATEMENT

Details about the construction of Know2BIO are provided in Appendices A and C. The source code and scripts for experiments are available at `https://anonymous.4open.science/r/Know2BIO/`. Experiments' procedures are provided at Section 4.2 and complete configuration files are provided at `https://anonymous.4open.science/r/Know2BIO/benchmark/configs`.

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

## A  KNOWLEDGE GRAPH SCHEMA

Figure 2 illustrates the organization of Know2BIO. White rectangles represent different source databases, within which the smaller rectangles with round corners represent different node types. The lines linking them represent the relationships between various node types and source databases. The figure shows the various biomedical relationships and prerequisite node identifier mappings/alignments needed to construct Know2BIO. The italicized text at the top of a database rectangle is the database name. The text without parentheses in a node type rectangles is the node type, and the text in parentheses is the identifier vocabulary used. [5]

Here we provide details on the biomedical categories and data sources in Table 3. Know2BIO integrates data of 11 biomedical types represented by 16 data types using 32 identifiers extracted from 30 sources. Biomedical types are anatomy, biological process, cellular component, compounds/drugs, disease, drug class, genes, molecular function, pathways, proteins, and reactions. Each biomedical type has at least one data type/identifier in Know2BIO. Due to unalignable/disjoint sets of pathways

---

[5]Although very recent versions of the data were used, the data used in this KG do not necessarily reflect the most current data from each source at the time of publication (e.g., PubMed, GO).

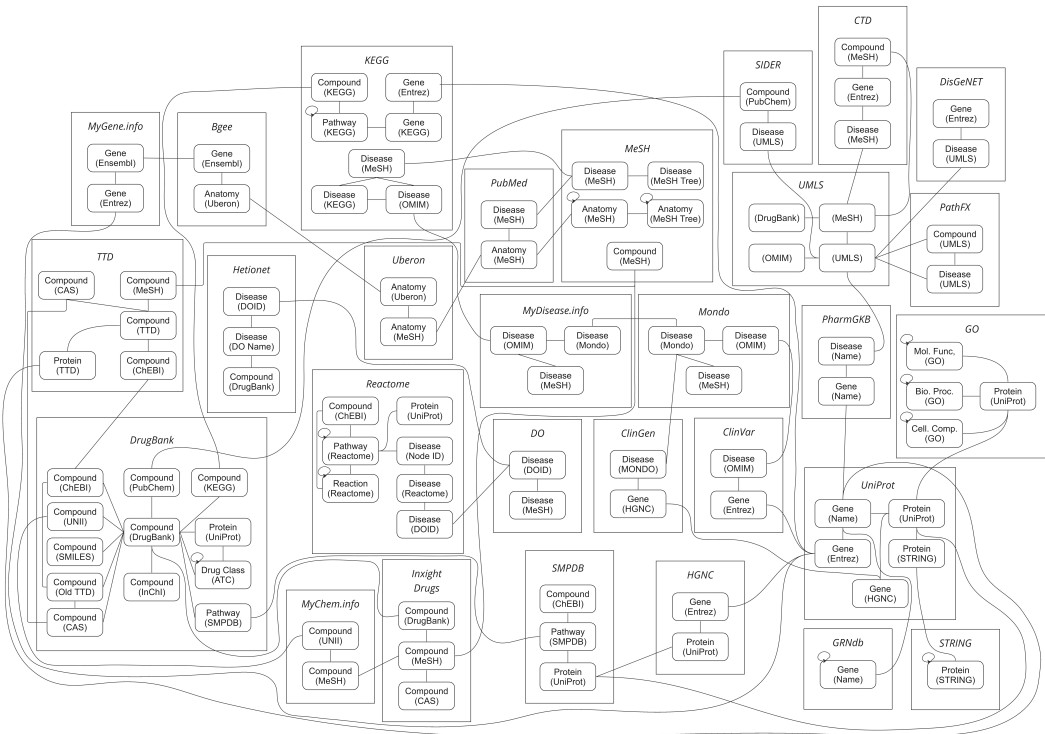

Figure 2: Database schema of Know2BIO.

across pathway databases, 3 pathway identifiers are used (Reactome, KEGG, SMPDB). Because we need to represent both the ontological structure of the anatomy data and MeSH disease, the anatomy and disease have 2 identifiers, one for unique MeSH IDs pointing to the potentially multiple MeSH tree numbers in the ontology; and the other for compounds due to incomplete alignment between DrugBank and MeSH identifiers. The remaining data types have 1 identifier to which all other identifiers are aligned.

The identifiers used include those of DrugBank, Medical Subject Headings IDs (MeSH), MeSH tree numbers, the old Therapeutic Target Database (TTD), the current TTD, PubChem Substance, PubChem Compound, Chemical Entities of Biological Interest (ChEBI), ChEMBLMendez et al. (2018), Simplified Molecular Input Line Entry System (SMILES), Unique Ingredient Identifier (UNII), International Chemical Identifier (InChI), Anatomical Therapeutic Chemical Classification System (ATC), Chemical Abstracts Service (CAS), Disease Ontology, Online Mendelian Inheritance in Man (OMIM), Monarch Disease Ontology (Mondo), Gene Ontology, Small Molecule Pathway Database (SMPDB), Reactome, Kyoto Encyclopedia of Genes and Genomes (KEGG), Bgee, Uberon, SIDER Mozzicato (2020), Comparative Toxicogenomics Database (CTD), PharmGKB, Search Tool for the Retrieval of Interacting Genes/Proteins (STRING), UniProt, Gene Regulatory Network database (GRNdb), HUGO Gene Nomenclature Committee (HGNC), and Entrez, and Unified Medical Language System (UMLS).

Table 11: Data Source for Know2BIO

| Data Source | License |
|---|---|
| AlphaFold Jumper et al. (2021); Váradi et al. (2021) | CC-BY 4.0 |
| Bgee Bastian et al. (2020) | CC0 |
| CTD Davis et al. (2022) | Custom[6] |
| ClinGen Rehm et al. (2015) | CC0[7] |
| ClinVar Landrum et al. (2019) | Custom[8] |
| DO Schriml et al. (2021) | CC0 |
| DisGeNET Piñero et al. (2021) | CC BY-NC-SA 4.0 |
| DrugBank Wishart et al. (2017) | CC BY-NC 4.0 International[9] |
| GO Carbon et al. (2020); Ashburner et al. (2000) | CC Attribution 4.0[10] Unported |
| GRNdb Fang et al. (2020) | Custom Fang et al. (2020)[11] |
| HGNC Seal et al. (2022) | CC0 |
| Hetionet Himmelstein et al. (2017) | CC0 |
| Inxight Drugs Siramshetty et al. (2021) | None provided |
| KEGG Kanehisa et al. (2022) | Custom[12] |
| MeSH Lipscomb (2000) | Custom[13] |
| Mondo Vasilevsky et al. (2022) | CC-BY 4.0 |
| MyChem.info Lelong et al. (2021) | Custom[14] |
| MyDisease.info Lelong et al. (2021) | Custom[15] |
| MyGene.info Lelong et al. (2021) | Custom[16] |
| PathFX Wilson et al. (2018) | CC0/CC-BY 4.0 |
| PharmGKB Gong et al. (2021) | CC-BY 4.0[17] |
| PubMed [18] | Custom[19] |
| Reactome Gillespie et al. (2021) | CC0 |
| SIDER Kuhn et al. (2015) | CC-BY-NC-SA 4.0 |
| SMPDB Jewison et al. (2013) | None provided |
| STRING Szklarczyk et al. (2018) | CC-BY |
| TTD Zhou et al. (2021) | None provided[20] |
| Uberon Haendel et al. (2014); Mungall et al. (2012) | CC-BY 3.0 |
| UMLS Bodenreider (2004) | Custom[21] |
| UniProt Bateman et al. (2022) | CC-BY 4.0 |

The data sources include various databases, knowledge bases, API services, and knowledge graphs: MyGene.info, MyChem.info, MyDisease.info, Bgee, KEGG, PubMed, MeSH, SIDER, UMLS,

---

[6] https://ctdbase.org/about/legal.jsp

[7] Its sources CGI & PharmGKB are CC0 https://clinicalgenome.org/tools/clingen-website/attribution/

[8] https://www.ncbi.nlm.nih.gov/clinvar/docs/maintenance_use/

[9] https://go.drugbank.com/about

[10] http://geneontology.org/docs/go-citation-policy/

[11] freely accessible for non-commercial use

[12] https://www.kegg.jp/kegg/legal.html

[13] https://www.nlm.nih.gov/databases/download/terms_and_conditions_mesh.html

[14] https://mychem.info/terms

[15] https://mychem.info/terms

[16] https://mygene.info/terms

[17] https://creativecommons.org/licenses/by-sa/4.0/

[18] https://pubmed.ncbi.nlm.nih.gov/

[19] "Terms and Condition" in https://ftp.ncbi.nlm.nih.gov/pubmed/baseline/README.txt

[20] https://db.idrblab.net/ttd/

[21] https://www.nlm.nih.gov/databases/umls.html, https://www.nlm.nih.gov/databases/umls.html

CTD, PathFX, DisGeNET, TTD, Hetionet, Uberon, Mondo, PharmGKB, DrugBank, Reactome, DO, ClinGen, ClinVar, UniProt, GO, STRING, InxightDrugs, SMPDB, HGNC, and GRNdb.

We used these for different edges: Bgee for gene-anatomy edges; CTD for compound-gene and gene-disease; ClinGen for gene-disease; ClinVar for gene-disease; Disease Ontology for disease-disease alignments; DisGeNET for gene-disease; DrugBank for compound-compound (interactions and alignment), protein-compound, and pathway-compound; Gene Ontology for GO term ontology edges of molecular function, biological process, and cellular component, as well as edges between the GO terms and proteins; GRNdb for transcription factor to regulon edges, i.e., protein-gene; HGNC for gene-protein; Hetionet for compound-disease; Inxight Drugs for compound-compound alignments; KEGG for compound-pathway, pathway-pathway, pathway-gene, and alignments for disease-disease and gene-gene; MeSH for disease-disease, anatomy-anatomy, and compound-compound alignments, as well as disease-disease and anatomy-anatomy ontology edges; Mondo for disease-disease alignments; MyChem.info for compound-compound alignments; MyDisease.info for compound-compound alignments; MyGene.info for gene-gene alignments; PathFX for compound-disease; PharmGKB for gene-disease; PubMed for disease-anatomy; Reactome for reaction-reaction, compound-reaction, pathway-reaction, pathway-pathway, disease-pathway, and pathway-pathway, as well as alignments for disease-disease; SIDER for compound-disease (i.e., side effect / adverse drug event); SMPDB for protein-pathway and compound-pathway; STRING for protein-protein; TTD for compound-compound and protein-protein alignments, as well as compound-protein; Uberon for anatomy-anatomy alignments; UMLS for disease-disease and compound-compound alignments; and UniProt for protein-protein and gene-gene alignments.

The way in which the data and the identifiers were mapped to each other and merged into the same node is shown in Figure 2 and in the source code on GitHub, with provided documentation in the notebooks and README file. Except for the additional 5 of the 16 main identifiers discussed above, all other identifiers were mapped/aligned (often circuitously) to the main identifier types. In Know2BIO, these entities/concepts are represented by a unique node, not duplicating for the different identifiers as this would be computationally counterproductive and not biomedically insightful.

Node feature data is also included. DNA sequences were obtained from Ensembl and UniProt. Protein sequences were obtained from UniProt. Compound sequences were obtained from DrugBank. Protein structures were obtained from EBI DeepMind. Natural language names were obtained from the nodes' respective data sources.

Graph benchmarks are often very large. Therefore, we follow the common graph benchmarking practice of subdividing the data to be benchmarked on multiple basic models. Here, we separately benchmark the ontology and instance views and then benchmark the whole dataset. Various toolkits have been developed to expedite the repetitive and time-consuming task of adapting models to datasets Han et al. (2018); Sadeghi et al. (2021); Cappelletti et al. (2021); Ali et al. (2020). We use the OpenKE Han et al. (2018) toolkit as it provides base models and tasks needed.

Below, we summarize the mapping process in more detail for the scripts that create the edge files / triple files[22]:

**anatomy_to_anatomy** The official xml file from MeSH was used to map anatomy MeSH IDs and MeSH tree numbers to each other, as well as MeSH tree numbers to each other to form the hierarchical relationships in the ontology. MeSH IDs were aligned to Uberon IDs via the official Uberon obo file (used in gene-to-anatomy)

**compound_to_compound** The compound_to_compound_alignment script aligned numerous compound identifiers in order to align DrugBank and MeSH IDs, two of the most prevalent IDs from data sources for different relationships in the scripts here. To produce this file, numerous resources were used to directly map DrugBank to MeSH IDs or to indirectly align the IDs (e.g., via DrugBank to UNII from DrugBank, then UNII to MeSH via MyChem.info). Resources include UMLS's MR-CONSO.RRF file, DrugBank, MeSH, MyChem.info, the NIH's Inxight Drugs, KEGG, and TTD. In other scripts, DrugBank and MeSH compounds are mapped to one another via this mapping file.

Compound interactions were extracted from DrugBank.

---

[22]https://anonymous.4open.science/r/Know2BIO/dataset/create_edge_files_utils

**compound_to_disease** The majority of the compound-treats-disease and compound-biomarker_of-disease edges were from the Comparative Toxicogenomics Database. Additional edges were from PathFX (i.e., from repoDB) and Hetionet (reviewed by 3 physicians).

**compound_to_drug_class** Mappings from compounds to drug classes (ATC) were provided by DrugBank.

**compound_to_gene** Mapping compound to gene largely relies on CTD, though some relationships come from KEGG. Like many other compound mappings, this relies on the DrugBank-to-MeSH alignments from compound_to_compound_alignment.

**compound_to_pathway** Mapping compounds to SMPDB pathways relies on DrugBank. Mapping compounds to Reactome pathways relies on Reactome, plus alignments to ChEBI compounds. Mapping compounds to KEGG pathways relies on KEGG.

**compound_to_protein** Most compound-to-protein relationships are from DrugBank. Some are taken from TTD, relying on mappings provided by TTD and aligning identifiers based on DrugBank- and TTD-provided identifiers.

**disease_to_disease** The official xml file from MeSH was used to map disease MeSH IDs and MeSH tree numbers to each other, as well as MeSH tree numbers to each other to form the hierarchical relationships in the ontology.

To measure disease similarity, edges were obtained from DisGeNET's curated data. The UMLS-to-MeSH alignment was used (from compound_to_compound_alignment).

Disease Ontology was used to align Disease Ontology to MeSH. Mondo and MyDisease.info were relied on to align Mondo to MeSH, DOID, OMIM, and UMLS. These alignments were used to align relationships from other scripts to the MeSH disease identifiers.

**compound_to_side_effect** Mappings from compounds to the side effects they are associated with were provided by SIDER. This required alignments from PubChem to DrugBank (provided by DrugBank) and UMLS to MeSH (provided in compound_to_compound_alignment.py).

**disease_to_anatomy** Disease and anatomy association mappings rely on MeSH for aligning the MeSH IDs and MeSH tree numbers and rely on the disease-anatomy coocurrences in PubMed articles' MeSH annotations.

**disease_to_pathway** KEGG was used to map KEGG pathways to disease. Reactome was used to map Reactome pathways to diseases, relying on the DOID-to-MeSH alignments for disease.

**gene_to_anatomy** Gene expression in anatomy was derived from Bgee. To align the Bgee-provided Ensembl gene IDs to Entrez, MyGene.info was used. To align the Bgee-provided Uberon anatomy IDs to MeSH, Uberon was used (see anatomy_to_anatomy)

**gene_to_disease** Virtually all gene-disease associations were obtained from DisGeNET's entire dataset. Additional associations—many of which were already present in DisGeNET—were obtained from ClinVar, ClinGen, and PharmGKB. (Users may be interested in only using the curated evidence from DisGeNET or increasing the confidence score threshold for DisGeNET gene-disease association. We chose a threshold of 0.06 based on what a lead DisGeNET author mentioned to the Hetionet creator in a forum.

**gene_to_protein** We relied on UniProt and HGNC to map proteins to the genes that encode them. Notably, there is a very large overlap between these sources ( 95%). HGNC currently broke, so only UniProt is being used.

**go_to_go** The source of the Gene Ontology ontologies is Gene Ontology itself.

**go_to_protein** The source of the mappings between proteins and their GO terms is Gene Ontology.

**pathway_to_pathway** The source of pathway hierarchy mappings for KEGG is KEGG and for Reactome is Reactome. (SMPDB does not have a hierarchy)

**protein_and_compound_to_reaction** The source of mappings from proteins and compounds to reactions is Reactome. This file relies on alignments from ChEBI to DrugBank.

**protein_and_gene_to_pathway** To map proteins and genes to pathways, KEGG was used for KEGG pathways (genes), Reactome for Reactome pathways (proteins and genes), and SMPDB for SMPDB pathways (proteins).

**protein_to_gene_ie_transcription_factor_edges** To map the proteins (i.e., transcription factors) to their targeted genes (i.e., the proteins that affect expression of particular genes), GRNdb's high confidence relationships virtually all derived from GTEx, were used. This also required aligning gene names to Entrez gene IDs through MyGene.info

**protein_to_protein** Protein-protein interactions (i.e., functional associations) were derived from STRING. To map the STRING protein identifiers to UniProt, the UniProt API was used. A confidence threshold of 0.7 was used. (Users may adjust this in the script)

**reaction_to_pathway** To map reactions to the pathways they participate in, Reactome was used.

**reaction_to_reaction** To map reactions to reactions that precede them, Reactome was used.

## B    KNOWLEDGE GRAPH MODELS BENCHMARKED IN EXPERIMENTS

The KG representation learning models used for experiments can be classified into five categories based on their mechanism (scoring function, etc.): translation-based models, bilinear models, neural network models, complex vector models, and hyperbolic space models. Generally, the neural network models' scoring functions are very flexible and can include various spatial transformations; while most translation-based and bilinear models are models in Euclidean space.

*Translation-based models*, also known as Trans-X models, conceptualize relations as translation operations on the representations of entities. For example, TransE perceives each relation type as a translation operator that moves from the head entity to the tail entity. The principle of this movement can be represented mathematically as $v_h + v_r \approx v_t$. TransE is particularly suited for capturing 1-to-1 relationships, where each head entity is linked to a maximum of 1 tail entity for a given relation type. Later, TransH, TransR, and TransD extended the core idea of translation-based representation.

*Bilinear models* such as DistMult represent each relation as a diagonal matrix, facilitating interactions between entity pairs. SimplE is an extension of DistMult, allowing for the learning of two dependent embeddings for each entity.

*Neural network models* leverage neural networks (e.g. convolutional neural networks) for knowledge graph embedding. ConvE and ConvKB are prime examples. ConvE employs a convolution layer directly on the 2D reshaping of the embeddings of the head entity and relation. ConvKB applies a convolution layer over embedding triples. Each of these triples is represented as a 3-column matrix, where each column vector represents one element of the triple.

*Complex vector* models use vectors from Complex or Euclidean space to expand their expressive capacity. Notable examples include ComplEx, RotatE, and AttE.

*Hyperbolic space models* take advantage of hyperbolic space's ability to represent hierarchical structures with minimal distortion. In Euclidean space, distances between points are measured using the Euclidean metric, which assumes a flat space. However, in hyperbolic space, distances are measured using the hyperbolic metric, which takes into account the negative curvature of the space. This property allows hyperbolic space models to capture long-range dependencies more efficiently than Euclidean space models. Models like RefH and AttH enhance the quality of KG embedding by incorporating hyperbolic geometry and attention mechanisms to model complex relational patterns.

## C    RELATION TABLE IN KNOW2BIO

Out of the 6.18 million edges, there are 108 unique edge types. While most edges (i.e., relations) are between only one pair of biomedical categories, some relations exist across multiple pairs (e.g., the -is_a- edge connects drug classes to drug classes, diseases to diseases, anatomies to anatomies, pathways to pathways, and GO terms to GO terms for the ontology edges). Detailed in Tables 13 & 14, there are 30 unique pairs of biomedical category nodes, with the number of unique relationships between each pair of biomedical categories and the names of relations between them. Compound-compound is the node pair with the highest number of relations, with over 2.9 million edges across

Table 12: Model categorization and scoring functions

| | Model | Scoring function $f(h, r, t)$ |
|---|---|---|
| **Translation** | TransE Bordes et al. (2013) | $-\|\mathbf{h} + \mathbf{r} - \mathbf{t}\|_{1/2}$ where $\mathbf{r} \in \mathbb{R}^k$ |
| | TransH Wang et al. (2014) | $-\|(\mathbf{I} - r_p r_p^{\top})\mathbf{h} + \mathbf{r} - (\mathbf{I} - r_p r_p^{\top})\mathbf{t}\|_{1/2}$ where $r_p, \mathbf{r} \in \mathbb{R}^k$, $\mathbf{I}$ denotes an identity matrix size $k \times k$ |
| | TransR Lin et al. (2015) | $-\|\mathbf{M}_r\mathbf{h} + \mathbf{r} - \mathbf{M}_r\mathbf{t}\|_{1/2}$ where $\mathbf{M}_r \in \mathbb{R}^{n \times k}$, $\mathbf{r} \in \mathbb{R}^n$ |
| | TransD Ji et al. (2015) | $-\|(\mathbf{I} + r_p h_p^{\top})\mathbf{h} + \mathbf{r} - (\mathbf{I} + r_p t_p^{\top})\mathbf{t}\|_{1/2}$ where $\mathbf{r}, r_p, h_p, t_p \in \mathbb{R}^k$ |
| **Bilinear** | DistMult Yang et al. (2014) | $\mathbf{h}^{\top}\mathbf{M}_r\mathbf{t}$ where $\mathbf{M}_r$ is a diagonal matrix $\in \mathbb{R}^{k \times k}$ |
| | SimplE Kazemi & Poole (2018) | $\frac{1}{2}(\mathbf{h_1}^{\top}\mathbf{M}_r\mathbf{t_2} + \mathbf{t_1}^{\top}\mathbf{M}_{r^{-1}}\mathbf{h_2})$ where $\mathbf{h_1}, \mathbf{h_2}, \mathbf{t_1}, \mathbf{t_2} \in \mathbb{R}^k$; $\mathbf{M}_r$ and $\mathbf{M}_{r^{-1}}$ are diagonal matrices $\in \mathbb{R}^{k \times k}$ |
| **Neural network** | NTN Socher et al. (2013) | $\mathbf{r}^{\top}\tanh(\mathbf{h}^{\top}\mathbf{M}_r\mathbf{t} + \mathbf{M}_{r,1}\mathbf{h} + \mathbf{M}_{r,2}\mathbf{t} + \mathbf{b}_r)$ where $\mathbf{r}, \mathbf{b}_r \in \mathbb{R}^n$; $\mathbf{M}_r \in \mathbb{R}^{k \times k \times n}$; $\mathbf{M}_{r,1}, \mathbf{M}_{r,2} \in \mathbb{R}^{n \times k}$ |
| | ER-MLP Dong et al. (2014) | $\text{sigmoid}(\mathbf{w}^{\top}\tanh(\mathbf{W} \cdot \text{concat}(\mathbf{h}, \mathbf{r}, \mathbf{t})))$ |
| | ConvE Dettmers et al. (2017) | $\mathbf{t}^{\top}\text{ReLU}\left(\mathbf{W} \cdot \text{vec}\left(\text{ReLU}\left(\text{concat}(\overline{\mathbf{h}}, \overline{\mathbf{r}}) * \mathbf{\Omega}\right)\right)\right)$ where $\overline{\mathbf{h}}$ and $\overline{\mathbf{r}}$ denote a 2D reshaping of $\mathbf{h}$ and $\mathbf{r}$, respectively |
| | ConvKB Nguyen et al. (2017) | $\mathbf{w}^{\top}\text{concat}\left(\text{ReLU}\left([\mathbf{h}, \mathbf{r}, \mathbf{t}] * \mathbf{\Omega}\right)\right)$ |
| **Complex** | ComplEx Trouillon et al. (2016) | $\text{Re}\left(c_h^{\top}\mathbf{C}_r\hat{c}_t\right)$ where $\text{Re}(c)$ denotes the real part of the complex value $c \in \mathbb{C}$; $c_h, c_t \in \mathbb{C}^k$; $\mathbf{C}_r \in \mathbb{C}^{k \times k}$ is a diagonal matrix; $\hat{c}_t$ is the conjugate of $c_t$ |
| | RotatE Sun et al. (2018) | $-\|c_h \circ c_r - c_t\|_{1/2}$ where $c_h, c_r, c_t \in \mathbb{C}^k$; $\circ$ denotes the element-wise product |
| **Hyperbolic** | MuRP Balazevic et al. (2019) | $-d_{\mathbb{B}}\left(\exp_0^c(\mathbf{R}\log_0^c(\mathbf{h})), \mathbf{t} \oplus_c \mathbf{r}\right)^2 + b_h + b_t$ where $\mathbf{h}, \mathbf{r}, \mathbf{t} \in \mathbb{B}_c^d, b_h, b_t \in \mathbb{R}$ |
| | RefH Chami et al. (2020) | $-d_{\mathbb{B}}^{c_r}\left(\mathbf{q}_{\text{Ref}}^H, \mathbf{e}_t^H\right)^2 + b_h + b_t$ where $\mathbf{h}, \mathbf{t} \in \mathbb{B}_c^d, b_h, b_t \in \mathbb{R}, \mathbf{r} \in \mathbb{B}_c^d, \mathbf{q}_{\text{Ref}}^H = \text{Ref}(\Theta_r)\mathbf{e}_h^H$ |
| | RotH Chami et al. (2020) | $-d_{\mathbb{B}}^{c_r}\left(\mathbf{q}_{\text{Rot}}^H, \mathbf{e}_t^H\right)^2 + b_h + b_t$ where $\mathbf{h}, \mathbf{t} \in \mathbb{B}_c^d, b_h, b_t \in \mathbb{R}, \mathbf{r} \in \mathbb{B}_c^d, \mathbf{q}_{\text{Rot}}^H = \text{Rot}(\Theta_r)\mathbf{e}_h^H$ |
| | AttH Chami et al. (2020) | $-d_{\mathbb{B}}^{c_r}\left(\text{Att}\left(\mathbf{q}_{\text{Rot}}^H, \mathbf{q}_{\text{Ref}}^H; \mathbf{a}_r\right) \oplus^{c_r} \mathbf{r}_r^H, \mathbf{e}_t^H\right)^2 + b_h + b_t$ where $\mathbf{h}, \mathbf{t} \in \mathbb{B}_c^d, b_h, b_t \in \mathbb{R}, \mathbf{r} \in \mathbb{B}_c^d$ |

two types of relations: '-is-' and '-interacts_with->' indicating an alignment between two identical drugs and interaction between two compounds, respectively. While most pairs of biomedical concepts consist of one or two types of relations, the pair with the largest number of relation types is between protein and compound with 51 different relations, shown separately in Table 14 for practical purposes. These relations describe specifically how a protein interacts with a compound.

## D  DATASET ACCESSIBILITY AND MAINTENANCE

The intended use of this dataset is as a general-use biomedical KG. We note that many other biomedical KGs were constructed with a single use-case in mind and were often assembled in a one-time effort and have not been updated continuously. Source codes used to generate and update this dataset as well as the accompanying software codes to process and model this KG are available at https://anonymous.4open.science/r/Know2BIO. Datasheet describing the dataset and accompanying metadata is also included in the GitHub repository. The licenses for all datasets are detailed in Table 11. We acknowledge that we bear responsibility in case of violation of license and rights for data included in our KG. We release the data available under the respective licenses of the data sources (See Table 9) license publicly; the remainder are available upon request with the appropriate easily-requestable academic credentials from DrugBank. Some resources require free accounts to access and use the data (e.g., UMLS). The source code to obtain the data is released under MIT license and the data are released under the respective license of the data sources. The dataset will be updated periodically as new biomedical knowledge are updated and made available. The dataset is currently not yet released and will be released upon acceptance of the manuscript, through the GitHub repository. The dataset is available in three formats: 1) as raw input files (.csv) detailing individually extracted biomedical knowledge via API and downloads. These files also include intermediate files for mapping between ontologies as well as node features (e.g., text descriptions, sequence data, structure data), and edge weights which were not included in the combined dataset as they were not included in the model evaluation. A folder also contains only the final edges to be used in the KG. 2) a combined KG following the head-relation-tail (h,r,t) convention, as a comma-separated text file. These KGs are released for the ontology view, instance view, and bridge view, as well as a combined whole KG. 3) To facilitate benchmark comparison between different KG embedding models, we also release the train, validation, and test split KGs. Long-term preservation of the dataset will be done through versioning as the data are updated and the source codes are run to construct the updated KG. The construction of this KG uses the API available through numerous APIs and biomedical research knowledge sources. Therefore, the source codes to construct the KG may deprecate when

Table 13: Unique Relations Between Entity Types [1]

| Head Type | Tail Type | # Type of Relations | # Triple | Relations |
|---|---|---|---|---|
| Gene | Compound | 2 | 546 | -decreases->, -increases-> |
| Disease | Pathway | 1 | 751 | -disease_involves-> |
| Pathway | Pathway | 2 | 3025 | -pathway_is_parent_of->, isa |
| Compound | Drug Class | 1 | 5152 | -is- |
| Drug Class | Drug Class | 1 | 5707 | isa |
| Anatomy | Anatomy | 2 | 6299 | -is-, isa |
| Cellular Component | Cellular Component | 1 | 6498 | isa |
| Compound | Reaction | 1 | 11934 | -participates_in-> |
| Molecular Function | Molecular Function | 1 | 13747 | isa |
| Reaction | Pathway | 1 | 14925 | -involved_in-> |
| Compound | Pathway | 3 | 17401 | -compound_participates_in->, -drug_participates_in_pathway->, -drug_participates_in-> |
| Gene | Protein | 1 | 21330 | -encodes-> |
| Biological Process | Biological Process | 4 | 64560 | -negatively_regulates->, isa, -positively_regulates->, -regulates-> |
| Compound | Disease | 1 | 67715 | -treats-> |
| Protein | Molecular Function | 4 | 72032 | NOT\|enables, NOT\|contributes_to, enables, contributes_to |
| Gene | Pathway | 1 | 80486 | -may_participate_in-> |
| Protein | Cellular Component | 8 | 89741 | is_active_in, colocalizes_with, NOT\|colocalizes_with, NOT\|part_of, NOT\|is_active_in, located_in, NOT\|located_in, part_of |
| Disease | Disease | 4 | 136406 | -diseases_share_variants-, -is-, isa, -diseases_share_genes- |
| Protein | Biological Process | 10 | 139399 | NOT\|acts_upstream_of_or_within_negative_effect, acts_upstream_of, acts_upstream_of_or_within_negative_effect, acts_upstream_of_positive_effect, acts_upstream_of_negative_effect, acts_upstream_of_or_within_positive_effect, acts_upstream_of_or_within, NOT\|involved_in, NOT\|acts_upstream_of_or_within, involved_in |
| Gene | Disease | 2 | 201336 | -not_associated_with-, -associated_with- |
| Protein | Reaction | 5 | 209254 | -output->, -entityFunctionalStatus->, -regulatedBy->, -input->, -catalystActivity-> |
| Gene | Anatomy | 2 | 217166 | -overexpressed_in->, -underexpressed_in-> |
| Protein | Protein | 1 | 245958 | -ppi- |
| Protein | Pathway | 2 | 350832 | -participates_in->, -may_participate_in-> |
| Compound | Gene | 6 | 487733 | increases, -decreases->, -associated_with->, -affects->, -increases->, decreases |
| Protein | Gene | 1 | 748831 | -transcription_factor_targets-> |
| Compound | Compound | 2 | 2902659 | -is-, -interacts_with-> |

these resources update their APIs. However the functionality will be restored upon the next update of the dataset.

Table 14: Unique Relations Between Entity Types [2]

| Head Type | Tail Type | # Type of Relations | # Triple | Relations |
|---|---|---|---|---|
| Drug | Protein | 51 | 59737 | -binder->, -inhibitor->, -translocation_inhibitor->, -drug_targets_protein->, -chelator->, -inhibitory_allosteric_modulator->, -inverse_agonist->, -allosteric_modulator->, -antagonist->, -unknown->, -product_of->, -inactivator->, -cofactor->, -regulator->, -chaperone->, -partial_antagonist->, -other/unknown->, -cleavage->, -inhibits_downstream_inflammation_cascades-> -neutralizer->, -gene_replacement->, -blocker->, -drug_uses_protein_as_carriers-, -partial_agonist->, -incorporation_into_and_destabilization->, -suppressor->, -drug_uses_protein_as_enzymes-, -drug_uses_protein_as_transporters-, -multitarget->, -potentiator->, -inducer->, -binding->, -degradation->, -stimulator->, -antisense_oligonucleotide->, -modulator->, -component_of->, -substrate->, -positive_allosteric_modulator->, -downregulator->, -weak_inhibitor->, -activator->, -other->, -stabilization->, -inhibition_of_synthesis->, -agonist->, -ligand->, -negative_modulator->, -antibody->, -oxidizer->, -nucleotide_exchange_blocker-> |

