# OpenReview forum: "Know2BIO: A Comprehensive Dual-View Benchmark for Evolving Biomedical Knowledge Graphs"
_ICLR.cc/2024/Conference — Submitted to ICLR 2024_

### Official Review · Reviewer_sXSH · 2023-10-19

**Soundness:** 3 good
**Presentation:** 3 good
**Contribution:** 1 poor
**Rating:** 1
**Confidence:** 3

**Summary:**

The paper discuss the building of clinical knowledge graph by aggregating several existing knowledge grpahs

**Strengths:**

-

**Weaknesses:**

I don't think the correct venue for this work, clinical/biology venues are a better fit

**Questions:**

Other than evaluation there are no ML aspects to this work, am I missing something?

---

> ### Author Response · Authors · 2023-11-20
> **Reply to Weakness 1**
>
> *I don't think the correct venue for this work, clinical/biology venues are a better fit*
>
> We respectfully disagree. Although our knowledge graph draws from the biomedical field, it can inspire new research in both biomedicine and knowledge graph representation learning fields. Our benchmark dataset represents a novel challenge for computer science researchers, as it represents real-world data obtained from many heterogeneous sources, within the biomedical field.
>
> Of note, none of the existing works so far show remarkable performance on Know2BIO, demonstrating a huge room for improvement in model development, which can inspire new advances in representation learning. Furthermore, our mechanism for updating and benchmarking as up-to-date as possible real-world data are concepts applicable to other benchmark datasets and research fields.

---

### Official Review · Reviewer_Jbrt · 2023-10-24

**Soundness:** 3 good
**Presentation:** 3 good
**Contribution:** 2 fair
**Rating:** 3
**Confidence:** 3

**Summary:**

This paper proposes a large-scale heterogeneous knowledge graph benchmark named Know2BIO in the biomedical field, integrated from 30 data sources. The graph includes multi-modal data, including text descriptions, sequences of proteins and compounds, structures of proteins and compounds, etc. The authors compared several triple-based knowledge base completion models on the graph.

**Strengths:**

- The paper proposes a large-scale, heterogeneous, automatically updated biomedical knowledge graph with several multi-modal attributes, including texts, sequences of proteins and compounds, and their structures.
- Several methods are evaluated and compared on the proposed graph.

**Weaknesses:**

- Although the proposed graph is large and periodical update is important and needs considerable effort, constructing knowledge graphs from existing databases is not novel and the difficulty in constructing such KGs is limited.
- The proposed graph has several multi-modal attributes, but they are not used in the evaluation and the compared methods are limited.
- The paper misses several related works on heterogeneous KG.
  - Wise et al., COVID-19 Knowledge Graph: Accelerating Information Retrieval and Discovery for Scientific Literature. In Proceedings of Knowledgeable NLP: the First Workshop on Integrating Structured Knowledge and Neural Networks for NLP, 2020.
  - Asada et al., Integrating heterogeneous knowledge graphs into drug–drug interaction extraction from the literature. Bioinformatics, 2023.

**Questions:**

See the weaknesses.

---

> ### Author Response · Authors · 2023-11-20
> **Reply to Weakness 1**
>
> *Although the proposed graph is large and periodical update is important and needs considerable effort, constructing knowledge graphs from existing databases is not novel and the difficulty in constructing such KGs is limited.*
>
> Thank you for bringing this up.
>
> We respectfully disagree that constructing biomedical knowledge graphs poses minimal challenges, particularly in the biomedical domain. To construct a quality knowledge graph, researchers must have the biological expertise to fully understand the relationships and entities, as well as any associated confidence metrics.
>
> Furthermore, proper mapping of diverse entities across multiple knowledge bases is not a trivial task. Finally, our knowledge graph is represented in a dual-view setting, which needs expertise to properly discern whether entities and/or relationships are ontological concepts rather than instantiated biomedical entities, for which not much prior work has been done.

---

> ### Author Response · Authors · 2023-11-20
> **Reply to Weakness 2**
>
> *The proposed graph has several multi-modal attributes, but they are not used in the evaluation and the compared methods are limited.*
>
> Thank you for the question.
>
> We highlight the significance of our benchmark dataset, which encompasses not only triple-based biomedical knowledge but also incorporates multi-modal attributes. In other words, Know2BIO emphasizes more on the dual-view feature (ontology and instance views) of biomedical KG as a novel benchmark.
>
> While our initial benchmark evaluations didn't encompass these multi-modal attributes, we have decided to include this valuable information with our release. Subsequently, we plan to conduct model evaluations that harness the potential of this data at a later stage, including BERT_KG and LLM models.

---

> ### Author Response · Authors · 2023-11-20
> **Reply to Weakness 3**
>
> *The paper misses several related works on heterogeneous KG.*
> - Wise et al., COVID-19 Knowledge Graph: Accelerating Information Retrieval and Discovery for Scientific Literature. In Proceedings of Knowledgeable NLP: the First Workshop on Integrating Structured Knowledge and Neural Networks for NLP, 2020.
> - Asada et al., Integrating heterogeneous knowledge graphs into drug–drug interaction extraction from the literature. Bioinformatics, 2023.
>
> Thank you for pointing this out. We apologize for this oversight. We have updated the paper to include these publications.

---

> > ### Comment · Reviewer_Jbrt · 2023-11-23
> >
> > The response did not resolve the weaknesses, so I will keep my score.

---

### Official Review · Reviewer_RhR2 · 2023-10-31

**Soundness:** 3 good
**Presentation:** 3 good
**Contribution:** 3 good
**Rating:** 6
**Confidence:** 5

**Summary:**

The authors in this paper propose Know2BIO, a heterogeneous KG benchmark for the medical domain. The KG integrated data from several sources and its very large in size. It consists of multi-modal data where the node features contain text descriptions, protein and compound sequences and structures. The authors evaluated KG representation models on the proposed KG to demonstrate its effectiveness as a benchmark. The authors have clearly laid out the limitations of the KG as well as the future work.

**Strengths:**

Some of the strengths of Know2BIO are:

1. Know2BIO is larger and contains information from several sources (30 sources).
2. It represents 11 biomedical categories and other edges types that are typically absent in other KGs.
3. It is robust, up-to-date and can be extensible.
4. With the information it contains, it can support several real-world learning tasks.

**Weaknesses:**

Using the data source code URL provided in the abstract of this paper, the authors’ identity is visible in the README. This goes against ICLR’s anonymity policies (authors also checked “ I certify that there is no URL (e.g., github page) that could be used to find authors' identity.” But should have verified this before submitting.

**Questions:**

1. Are the edges for each of the categories homogeneous in nature?
2. Can users easily query this KG? If yes, will the authors be extending support to using LLMs on this?

---

> ### Author Response · Authors · 2023-11-20
> **Reply to Question 1**
>
> Thank you for the question.
>
> Yes, the edges for each category are homogenous in nature. In total, there are 108 unique edge types, across 11 biomedical categories. In the Appendix, the final table(s) illustrate the number of relations between pairs of biomedical categories, with the most relationships between drug and protein nodes.

---

> ### Author Response · Authors · 2023-11-20
> **Reply to Question 2**
>
> Thank you for your constructive suggestion.
>
> - Querying the KG is easy for a 1-hop query: simply use grep to find the relationship/node of interest.
>
> - For k-hop queries, we recommend using a graph database management system, such as Neo4j. We have included code to load the resulting knowledge graph into Neo4j and query using its Cypher query language, with an example query. We have uploaded the codebase to deploy Neo4j on Know2BIO to the repository.
>
> - Support for integrating this knowledge graph with LLMs will be explored in a subsequent publication, currently in the works (LLaMA2).

---

> > ### Author Response · Authors · 2023-11-23
> > **Reply to Question 2**
> >
> > The code can be found here: https://anonymous.4open.science/r/Know2BIO/dataset/neo4j_visualization

---

### Official Review · Reviewer_EnCN · 2023-11-01

**Soundness:** 2 fair
**Presentation:** 2 fair
**Contribution:** 2 fair
**Rating:** 5
**Confidence:** 4

**Summary:**

Know2BIO is a biomedical Knowledge Graph that integrates multiple knowledge bases to create a big heterogenous KG with 219K nodes and 6.9M relationships.

**Strengths:**

The authors have done a good job in explaining the process of creating the knowledge graph for biomedical domain.

**Weaknesses:**

1. It would have been better if some examples were present that would explain the integration of specific similar entities with different names/representation in two KGs, and steps of resolving those.
2. Future work should provide concrete methods that could be tried.

**Questions:**

1. How did you integrate multiple knowledge graphs with different ontology?
2. What techniques did you use to perform entity resolution, entity disambiguation?
3. What challenges did you face while merging the different KGs?
4. Since you have mentioned the use of 30 KBs for your KG construction, how did you select these? What about other KGs present? Is there any significant KB not utilized?

---

> ### Author Response · Authors · 2023-11-20
> **Re: Weakness #1, Question 1, Question 2**
>
> Entity alignment was performed by identifier mapping. In other words, we relied on the entity mappings provided by the knowledge bases we used, not on a computational method that would be less reliable such as fuzzy matching or embedding-based approaches. Once we identify the relevant data sources, the process to align the entities is somewhat straightforward.
>
> An overview of the mappings from each data source is shown in the Appendix Figure 2. An example of the mapping process is described in the beginning of section 3.1. An overview of the mappings for each biomedical category is shown in the Appendix, starting at the bottom of page 21. The organized code to perform the mappings is provided in the GitHub repository under Know2BIO/dataset/create_edge_files_utils which we provide as a footnote in a few locations throughout the paper.
>
> Given this explanation, does the initial weakness still remain?

---

> ### Author Response · Authors · 2023-11-20
> **Re: Question 3, Question 4**
>
> identify the relevant data sources, the process to align the entities is somewhat straightforward.
> However, it was arduous to curate the proper data sources that provide the mappings (e.g., to find a path of “is_a” relationships between the head and tail entities from two data sources). We had certain biomedical relationships we were interested in (e.g.,  gene-disease, protein-function, or drug-side effect), based on our domain knowledge (our data curation team has backgrounds in biological sciences.) To find data sources that provide this information, we searched online, looked into data sources used by other well-known biomedical knowledge graphs, and investigated data sources we had known previously. Through this process, we became very familiar with the available resources and which ones were commonly used and reputable. To choose the main identifier to use for each data type (e.g., UniProt ID for all protein nodes), we depended on the identifier mappings that were available and used the most. For example, UniProt IDs are used almost universally, though some data sources have their IDs (e.g., STRING, TTD). Therefore, we chose to use UniProt and map the less commonly used IDs to UniProt. To reiterate, the decisions about biomedical data types were informed by our biomedical understanding and data availability; the decisions about the identifier mappings were informed by our understanding of and wrangling with the data sources.
>
> There are other KBs with additional data not contained in our 30 data sources which could be useful (e.g., clue.io). We are considering supplementing the KG with this information. However, the data is currently quite extensive, both in terms of biomedical information and data sources, and includes numerous major biomedical knowledge bases.
>
> Does this answer clarify our process or do the questions remain? We would be happy to provide more information and update the paper accordingly if needed.

---

> ### Author Response · Authors · 2023-11-21
> **Re: Weakness 2**
>
> Here is our new "Future Works" section with more concrete methods that could be tried:
>
> Future benchmarking work on Know2BIO can test various models in graph learning
> (e.g., KG embeddings, GNNs), multi-modal models (e.g., text and graph embedding models), and
> training strategies (e.g., parameter initialization, curriculum learning). Multi-view graph learning
> approaches can take advantage of the separate instance and ontology views. Multi-modal models
> can use the non-KG data modalities to further enrich the KG with a wider array of information. To
> elaborate, the non-KG data modalities can be extracted (e.g., by language models, graph learning
> models) and integrated as node information into knowledge graph learning models via different
> feature fusion approaches (e.g., early, mixed, late fusion). Some feature fusion approaches can also
> double as a pre-training parameter initialization strategy, one of many initialization strategies to
> test. The edge weights can be utilized by graph learning approaches that incorporate weights into
> the training objective. Edge weights can also be used for curriculum learning. This will enable
> researchers to access a more extensive and holistic view of biomedical knowledge, providing a wider
> range of benchmarking tasks and challenges.
>
> Additionally, other emerging technologies can take advantage of Know2BIO. In particular, large
> language models (LLMs) can leverage the up-to-date information and structure of Know2BIO. This
> can facilitate reasoning and mitigate hallucination for the LLM, both for predictive analyses (e.g.,
> in-context learning) and retrieval augmented generation (e.g., multi-hop question answering). We
> plan to explore each of these topics in future works.

---

### Meta-Review · Area_Chair_qMxc · 2023-12-23

**Metareview:**

The submission presents a comprehensive and potentially useful biomedical knowledge graph. The breadth of data integration and the potential applicability of the work are acknowledged and appreciated. However, there are some points needed for improvement. The lack of novelty in constructing knowledge graphs from existing databases limits its distinction from prior works. Additionally, the underutilization of the multi-modal data in evaluations and insufficient methodological details on entity integration and resolution in the KG lessens its practical impact. Unfortunately, the paper does not show sufficient strength to meet the acceptance.

**Justification For Why Not Higher Score:**

The methodological novelty is not strong enough.

**Justification For Why Not Lower Score:**

The paper has the potential usefulness for biomedical knowledge graph.

---

### Decision · Program_Chairs · 2024-01-16

Reject